# Kynurenines and Mitochondrial Disturbances in Multiple Sclerosis

**DOI:** 10.3390/ijms26115098

**Published:** 2025-05-26

**Authors:** Daniel Pukoli, László Vécsei

**Affiliations:** 1Department of Neurology, Esztergomi Vaszary Kolos Hospital, H-2500 Esztergom, Hungary; pukoli.daniel@med.u-szeged.hu; 2Department of Neurology, Faculty of Medicine, University of Szeged, H-6725 Szeged, Hungary; 3HUN-REN-SZTE Neuroscience Research Group, Danube Neuroscience Research Laboratory, Hungarian Research Network, University of Szeged (HUN-REN-SZTE), H-6725 Szeged, Hungary

**Keywords:** multiple sclerosis, kynurenine pathway, neurodegeneration, inflammation, mitochondria, glutamate excitotoxicity

## Abstract

Multiple sclerosis (MS) is a chronic autoimmune disease characterised by inflammation, demyelination, and neurodegeneration within the central nervous system. The pathogenesis of MS involves an immune-mediated attack on myelin and neurons, accompanied by blood–brain barrier dysfunction and chronic CNS inflammation. Central to MS pathology is dysregulation of the kynurenine pathway, which metabolises tryptophan into neuroactive compounds. Kynurenine pathway (KP) activation, driven by inflammatory cytokines, leads to the production of both neuroprotective (e.g., kynurenic acid, KYNA) and neurotoxic (e.g., quinolinic acid, QUIN) metabolites. Imbalance between these metabolites, particularly increased QUIN production, exacerbates glutamate excitotoxicity, oxidative stress, and mitochondrial dysfunction, contributing to neuronal and oligodendrocyte damage. Mitochondrial dysfunction plays a critical role in the pathophysiology of MS, exacerbating neurodegeneration through impaired energy metabolism and oxidative stress. This review integrates the current understanding of KP dysregulation in multiple sclerosis across disease stages. In RRMS, heightened KP activity correlates with inflammation and neuroprotection attempts through increased KYNA production. In contrast, SPMS and PPMS are associated with a shift towards a more neurotoxic KP profile, marked by elevated QUIN levels and reduced KYNA, exacerbating neurodegeneration and disability progression. Understanding these mechanisms offers insights into potential biomarkers and therapeutic targets for MS, emphasising the need for strategies to rebalance KP metabolism and mitigate neurotoxicity in progressive disease stages.

## 1. Introduction

Multiple sclerosis (MS) is a chronic autoimmune demyelinating disease of the central nervous system (CNS) that is a leading cause of neurological disability in young adults. MS is characterised by recurrent episodes of inflammation leading to focal demyelinating lesions and, over time, axonal injury and neurodegeneration [1]. Clinically, MS manifests in different forms: most patients initially have relapsing-remitting MS (RRMS), defined by acute attacks followed by remissions, but within two decades, the majority progress to secondary progressive MS (SPMS), a phase of gradual neurological decline without clear remissions. A smaller subset (about 10–15%) have primary progressive MS (PPMS) from the onset, characterised by a steady functional decline from the beginning [2]. Despite the heterogeneity of the clinical course, the underlying pathology involves an immune-mediated attack on CNS myelin and neurons, disruption of the blood–brain barrier, and activation of resident glia, culminating in multifocal inflammation, demyelination, and loss of neurons and glial cells [3].

MS pathogenesis is driven by an interplay between autoimmune inflammation and degenerative processes. In early relapsing stages, peripheral T cells and monocytes infiltrate the CNS and initiate demyelinating lesions, whereas in progressive stages, the inflammation becomes more compartmentalised behind a closed blood–brain barrier and is sustained by CNS-resident microglia, astrocytes, and chronically activated immune cells. This prolonged inflammatory milieu not only damages myelin but also imposes significant metabolic stress on neurons and glia. Notably, it is now recognised that neurodegeneration in MS is not solely the consequence of relapse-associated tissue injury. A substantial component of irreversible disability accumulates through progression independent of relapse activity (PIRA), a process increasingly linked to chronic, subclinical inflammation and mitochondrial dysfunction in the CNS. In this context, mitochondrial damage is not just a downstream effect, but a central contributor to disease progression. Numerous pro-inflammatory cytokines (e.g., interferon-gamma (IFN-γ) and tumour necrosis factor-alpha (TNF-α), IL-1β, IL-17) are elevated in MS lesions and cerebrospinal fluid (CSF) and can induce the kynurenine pathway (KP) of tryptophan metabolism in immune and glial cells. The KP produces several bioactive metabolites (“kynurenines”) that have profound effects on neurons and immune regulation, including both neuroprotective and neurotoxic compounds [3]. Meanwhile, chronic inflammation and demyelination also disrupt axonal energy homeostasis and mitochondrial function. Defects in the mitochondrial respiratory chain and impaired mitochondrial transport have been documented in MS lesions, creating an energy imbalance that fuels progressive neurodegeneration. The accumulation of oxidative stress and mitochondrial injury—particularly in the context of PIRA—reflects a self-sustaining pathological loop that drives CNS atrophy and clinical worsening even in the absence of relapses [4,5]. Recent research has shown that these two factors—kynurenine pathway dysregulation and mitochondrial dysfunction—are closely linked and contribute to the pathophysiology of MS. Kynurenine pathway metabolites and mitochondrial dysfunction form a pathogenic nexus linking the inflammatory and neurodegenerative arms of MS. Understanding these links is of great research interest as they offer potential explanations for the progressive neurodegeneration seen in MS and suggest novel biomarkers and therapeutic targets.

In this article, we provide a comprehensive overview of the kynurenine pathway and its metabolites, and how their perturbation contributes to MS pathomechanisms alongside mitochondrial dysfunction. We also discuss emerging therapeutic approaches aimed at modulating the kynurenine pathway or ameliorating mitochondrial dysfunction in MS. By reviewing pharmacological interventions such as the investigational drug laquinimod and repurposed drugs such as alpha-lipoic acid and metformin, we highlight the prospects of targeting these pathways to mitigate inflammation-induced neurodegeneration in MS. Our aim is to elucidate the causal and mechanistic links between KP metabolites and mitochondrial dysfunction in MS, and to highlight their importance both as biomarkers of disease activity and as promising avenues for neuroprotective therapy.

## 2. Overview of the Kynurenine Pathway

### 2.1. Tryptophan Metabolism and Kynurenine Pathway

Tryptophan (TRP) is an essential amino acid that is catabolised in the body via two main pathways: the methoxyindole (serotonin/melatonin) pathway and the kynurenine pathway. In mammals, the KP is the predominant pathway of TRP catabolism, accounting for approximately 95% of TRP degradation, with only ~5% being converted to serotonin and related metabolites [6]. The KP is active in both peripheral tissues and the CNS, where it plays a critical role in immune cell metabolism and in the generation of nicotinamide adenine dinucleotide (NAD^+^) for cellular energy homeostasis [7]. The first and rate-limiting step of the KP is the oxidative cleavage of the indole ring of TRP to form N-formyl-L-kynurenine, a reaction catalysed by either indoleamine 2,3-dioxygenase (IDO) or tryptophan 2,3-dioxygenase (TDO). TDO is largely expressed in the liver and regulates systemic TRP levels, whereas IDO (with two isoforms, IDO-1 and IDO-2) is widely expressed in extrahepatic tissues, including immune cells, microglia, and dendritic cells [8]. IDO is highly inducible by pro-inflammatory signals (in particular IFN-γ and other cytokines), and thus, IDO is upregulated during immune activation, shunting TRP into the KP. This has important immunoregulatory consequences: by depleting local TRP and producing downstream metabolites, IDO can suppress T-cell proliferation and modulate immune responses [9]. Indeed, a high blood kynurenine/tryptophan ratio is a recognised marker of IDO activation in chronic inflammation and correlates with disease activity in several conditions.

### 2.2. Key Enzymes and Metabolites

The product of the IDO/TDO reaction, L-kynurenine (KYN), is the central junction of the KP (see Figure 1). Kynurenine can be metabolised by several enzymatic pathways: transamination by kynurenine aminotransferases (KATs) produces kynurenic acid (KYNA), whereas hydroxylation by kynurenine-3-monooxygenase (KMO) yields 3-hydroxykynurenine (3-HK). A third minor pathway via kynureninase can convert KYN directly to anthranilic acid, although this pathway is less prominent in the brain. The fate of KYN depends largely on the cell type. Specifically, astrocytes express KAT enzymes but lack KMO, thereby channelling KYN towards KYNA production, whereas activated microglia and infiltrating macrophages express the full complement of KP enzymes (including KMO) needed to produce 3-HK and its downstream metabolites. In contrast, oligodendrocytes do not significantly express IDO/TDO and therefore cannot initiate TRP metabolism or produce quinolinic acid (QUIN); they rely on the uptake of external KP intermediates. This compartmentalisation means that under inflammatory conditions, microglia and macrophages tend to produce neurotoxic KP metabolites (via the KMO branch), whereas astrocytes produce neuroprotective KYNA via the KAT pathway [9].

Each KP metabolite has different neuroactive properties. KYNA is primarily neuroprotective, acting as an antagonist at NMDA receptors, attenuating excitotoxicity and oxidative stress [7]. Although KYNA has also been reported to antagonise α7-nicotinic acetylcholine receptors, this interaction remains controversial. Several studies have failed to replicate this effect, suggesting that the primary actions of KYNA are mediated through glutamatergic and other non-cholinergic pathways [10]. In contrast, QUIN, a potent NMDA receptor agonist, promotes excitotoxic neuronal and oligodendrocyte damage when excessively accumulated in the CNS. Similarly, 3-HK has moderate neurotoxicity through oxidative stress mechanisms. Another intermediate, 3-hydroxyanthranilic acid (3-HAA), has dual effects—it acts as a pro-oxidant at low concentrations, but also has antioxidant properties. Further downstream, picolinic acid (PIC) is generally considered to be neuroprotective, with metal chelating and immunomodulatory effects. The end product of the KP, NAD^+^, is critical for energy metabolism and DNA repair, underlining the importance of the pathway in cellular homeostasis [7,8,9]. Balanced KP metabolism is essential; a shift towards excessive QUIN and 3-HK production can trigger neurodegeneration, whereas favouring KYNA or PIC formation provides neuroprotection.

In summary, activation of the KP results in a mixture of neuroactive compounds: KYNA and PIC, which are largely neuroprotective, versus 3-HK and QUIN, which are neurotoxic (especially in excess), with 3-HAA having context-dependent redox effects. The balance between these metabolites—often summarised as the KYNA/QUIN ratio—can significantly influence neuronal viability. A shift in KP towards the QUIN-producing branch (e.g., due to increased KMO activity or reduced KAT activity) is associated with increased excitotoxic and oxidative injury in the brain. In contrast, favouring KYNA production can attenuate glutamate toxicity and oxidative stress.

## 3. Roles of the Kynurenine Pathway in MS Pathophysiology

The KP sits at the intersection of the immune and nervous systems, and its dysregulation has direct implications for MS. In the inflammatory milieu of MS, elevated cytokines (IFN-γ, TNF-α, etc.) drive IDO-1 expression and KP activation within CNS immune cells. Alterations in KP metabolism are thought to play a dual role in MS: on the one hand, KP activation serves as a mechanism to regulate immune responses (through TRP depletion and the generation of immunomodulatory kynurenines), but on the other hand, excessive or imbalanced KP activity creates a neurotoxic environment in the CNS. In the following sections, we will examine how the KP changes during different phases of MS and how these changes contribute to disease pathology.

### 3.1. Pathogenic Roles of Individual KP Metabolites in MS

As illustrated by the diverse and sometimes opposing effects of individual metabolites (Table 1), the dysregulation of kynurenine pathway metabolites in multiple sclerosis plays a pivotal role in driving disease progression rather than being a mere bystander phenomenon.

#### 3.1.1. Quinolinic Acid (QUIN)

Specifically, QUIN, a potent neurotoxin closely linked to MS lesion pathology, is predominantly synthesised by activated microglia and infiltrating macrophages equipped with the necessary enzymatic machinery (including IDO and KMO). These cells primarily produce QUIN within active MS lesions, particularly at the periphery of demyelinating plaques where inflammation is most pronounced [11,12,13].

QUIN exerts its damaging effects through several mechanisms:1.Excitotoxicity: QUIN acts as an NMDA receptor agonist, directly triggering excitotoxicity in neurons and oligodendrocytes, a process known as endogenous excitotoxicity [14]. Pathologically elevated QUIN levels, reaching micromolar concentrations within MS lesions, excessively activate neuronal NMDA receptors. QUIN-induced NMDA receptor activation triggers sustained neuronal depolarization and calcium overload, causing excitotoxic injury and contributing directly to axonal transections observed in acute MS lesions. This calcium overload initiates detrimental cascades, including the activation of calpains and caspases, generation of reactive nitrogen and oxygen species, mitochondrial dysfunction, and ultimately neuronal apoptosis or necrosis [9,15,16]. Because QUIN selectively overstimulates the NMDA receptor subunits NR2A/NR2B, neurons in the cortex and spinal cord that abundantly express these subunits are particularly vulnerable to excitotoxic damage. This selective susceptibility may explain the pronounced neuronal degeneration observed in MS lesions [9,10,15,17]. As a direct consequence, pharmacological blockade of NMDA receptor activation with specific antagonists effectively prevents QUIN-induced neuronal death, providing strong evidence that excitotoxicity is the primary mechanism underlying QUIN-mediated neurodegeneration [9,16].

In contrast, oligodendrocytes express relatively low levels of NMDA receptors, rendering them less directly susceptible to QUIN-induced excitotoxicity. Instead, QUIN indirectly affects these cells by stimulating excessive neuronal glutamate release, leading to increased extracellular glutamate concentrations. This secondary excitotoxicity damages oligodendrocytes and may also impact pre-myelinating cells, which, despite their limited NMDA receptor expression, remain vulnerable to glutamate-mediated toxicity [9,18].

2.Glutamate dysregulation: QUIN promotes excessive neuronal glutamate release and inhibits astrocytic glutamate uptake, specifically by impairing glutamate transporters and glutamine synthetase [19,20]. This perturbation leads to elevated ambient glutamate levels, creating an excitotoxic environment detrimental to oligodendrocytes and neurons. The resulting glutamate excess in MS white matter directly correlates with oligodendrocyte death and subsequent demyelination [21,22,23].3.Oxidative damage: QUIN acts as a potent pro-oxidant, catalysing the formation of hydroxyl radicals through metal chelation (e.g., Fe^2+^) and subsequent Fenton reactions [12]. By interacting synergistically with reactive oxygen species in the mitochondria, QUIN exacerbates lipid peroxidation and cellular energy depletion [24,25,26]. Exposure to QUIN leads to the formation of toxic lipid peroxidation products such as malondialdehyde and 4-hydroxynonenal (4-HNE) in oligodendrocyte membranes, which directly contribute to myelin destruction [27,28]. In addition, QUIN induces inducible nitric oxide synthase (iNOS) in microglia, which increases peroxynitrite (ONOO-) formation, further exacerbating the oxidative DNA and protein damage observed in MS lesions [29,30].4.Energy failure: Although QUIN normally serves as a precursor for NAD^+^ synthesis, excessive accumulation paradoxically leads to energy failure [31]. Intracellular QUIN inhibits mitochondrial respiratory complexes and induces opening of the mitochondrial permeability transition pore (PTP) by calcium overload [32,33]. It also consumes NAD^+^ in futile pathways if not effectively rescued to NAD^+^, thereby depleting cellular energy reserves [34].5.Cytoskeletal disruption: Recent evidence highlights the role of QUIN in aberrant phosphorylation of neuronal cytoskeletal proteins. QUIN induces hyperphosphorylation of tau proteins through phosphatase inhibition and NMDA receptor-mediated kinase activation, leading to microtubule destabilisation, aggregate formation, impaired axonal transport, and neuronal dysfunction. Abnormal tau phosphorylation correlates strongly with axonal degeneration in MS lesions. QUIN also promotes neurofilament phosphorylation, further disrupting neuronal integrity and contributing to the characteristic axonal swelling seen in MS plaques [9].

Taken together, these mechanisms highlight the central role of QUIN in linking neuroinflammation to neurodegeneration in MS. Clinically, elevated levels of QUIN correlate closely with disease progression [18] and axonal loss [35]. Experimental models support this notion: exposure to QUIN in cultured neurons or oligodendrocytes reproduces the hallmark pathological features of MS, including demyelination and cell death [36]. In addition, therapeutic inhibition of QUIN production using KMO or IDO inhibitors in experimental autoimmune encephalomyelitis (EAE) models attenuates disease severity [37]. Thus, QUIN represents an important therapeutic target, prompting ongoing research into KP enzyme inhibitors aimed at reducing QUIN-mediated neurotoxicity in MS.

#### 3.1.2. Kynurenic Acid (KYNA)

While QUIN promotes neurotoxicity in MS, KYNA often acts as a counterbalancing neuroprotective agent. As an endogenous antagonist of ionotropic glutamate receptors, KYNA broadly inhibits excitatory neurotransmission, primarily by attenuating NMDA receptor activity, with additional weak blockades of AMPA/kainate receptors [38,39,40]. At physiological nanomolar concentrations in the brain, KYNA regulates glutamate signalling and exerts neuroprotective effects by attenuating excitotoxicity and potentially scavenging free radicals.

KYNA’s protective mechanism in MS is primarily mediated by its ability to prevent excessive NMDA receptor activation, thereby reducing intracellular Ca^2+^ influx and protecting neurons from QUIN- and glutamate-induced injury. In addition, by antagonising α7 nicotinic receptors on presynaptic terminals, KYNA suppresses glutamate release, thereby dampening excitatory signalling under inflammatory conditions [9]. Interestingly, KYNA exhibits a Janus-faced effect in vitro: while it is a known NMDA receptor antagonist at micromolar concentrations, electrophysiological studies in rat hippocampal slices have shown that it can facilitate excitatory postsynaptic potentials at nanomolar concentrations [41]. This duality suggests that KYNA exerts complex neuromodulatory effects depending on its concentration and synaptic context, possibly via interactions with different receptor subtypes or modulatory sites. This mechanism is particularly relevant in MS, where synaptic hyperexcitability and glutamate release from activated immune cells contribute to neuronal damage. In addition, KYNA and its analogues may neutralise reactive species such as peroxynitrite, thereby exerting an antioxidant effect [42]. Notably, higher KYNA levels are generally associated with reduced neuronal loss in MS lesions. Interestingly, KYNA levels are highest in PPMS [43], a form of the disease characterised by slower, chronic neurodegeneration, suggesting that the CNS attempts to compensate for the lack of relapses by upregulating KYNA synthesis. However, despite the neuroprotective potential of KYNA, its levels in MS are often insufficient to counteract the overwhelming neurotoxic stimuli. Astrocytes, the primary source of KYNA, are reactive and present in MS plaques, but their KYNA production may be impaired due to reduced KAT enzyme expression [30]. This imbalance has stimulated interest in therapeutic strategies aimed at increasing KYNA or its analogues to restore neuroprotection.

#### 3.1.3. 3-Hydroxykynurenine (3-HK)

Other KP intermediates also contribute to MS pathogenesis. 3-hydroxykynurenine, produced by microglial KMO, is highly reactive. In MS lesions, 3-HK can damage cells by oxidative mechanisms—it readily undergoes auto-oxidation to produce hydrogen peroxide and hydroxyl radicals. Oligodendrocytes are particularly susceptible to 3-HK-induced oxidative apoptosis due to their low antioxidant defences [9]. The combination of 3-HK and QUIN appears to be particularly damaging: while either metabolite alone may cause limited damage at moderate concentrations, together they act synergistically to kill neurons. This synergy has been demonstrated in vivo: co-injection of subtoxic doses of QUIN and 3-HK into the rat brain resulted in lesions significantly larger than those caused by either compound alone [44]. Such findings suggest that in MS, the simultaneous presence of high levels of QUIN and 3-HK in active lesions creates a “toxic cocktail” that amplifies oxidative excitotoxic damage. Therefore, controlling 3-HK (via KMO inhibition) may not only reduce QUIN production, but also remove a direct source of ROS. Recent studies have further elucidated the role of KMO in regulating this balance, highlighting its potential as a therapeutic target in neurodegenerative diseases, including MS [45].

The kynurenine pathway also plays a critical role in immune regulation in MS (see review by Pukoli et al. [9]). Both KYN and its downstream metabolites can modulate immune cell function. KYN itself, at high concentrations, can inhibit T-cell proliferation and induce T-cell apoptosis. It is also noteworthy that IDO-driven TRP depletion may have an immunoregulatory role—in other contexts, IDO activity helps suppress T cell proliferation and promotes the generation of regulatory T cells via the aryl hydrocarbon receptor (AHR) pathway.

#### 3.1.4. Picolinic Acid (PIC)

Picolinic acid is a downstream metabolite of the kynurenine pathway, synthesised from 2-amino-3-carboxymuconate-6-semialdehyde via the enzyme 2-amino-3-carboxymuconate-6-semialdehyde decarboxylase (ACMSD) [46]. Unlike its neurotoxic counterpart QUIN, PIC has neuroprotective and immunomodulatory properties, suggesting a potential compensatory role in the pathogenesis of MS. PIC acts as a potent metal chelator, binding divalent metal ions such as iron and zinc, thereby alleviating metal-induced oxidative stress—a contributing factor in MS pathogenesis. In vitro studies indicate that PIC, at concentrations ranging from 1 to 4 mM in supernatants, can enhance macrophage effector functions by increasing IFN-γ-dependent nitric oxide synthase gene expression and inducing macrophage inflammatory proteins (MIP) 1α and 1β. Although the exact mechanism underlying the synergy between PIC and IFN-γ remains unclear, the PIC-induced upregulation of MIP-1α and β is thought to involve an iron chelation-dependent pathway [47]. In addition, PIC antagonises the excitotoxic effects of QUIN, preventing calcium influx and subsequent neuronal apoptosis [48]. Clinical studies have reported altered PIC levels in MS patients, particularly during active disease phases, suggesting a deficiency of this protective metabolite. Decreased PIC levels correlate with increased disease severity and progression, suggesting its potential as a biomarker for disease monitoring [18,43]. Increasing PIC synthesis or mimicking its activity is a promising therapeutic approach. Upregulation of ACMSD activity could shift KP metabolism towards increased PIC production, potentially restoring the balance between neuroprotective and neurotoxic metabolites. The development of PIC analogues that retain its protective properties may offer therapeutic benefits without the complexity of modulating endogenous pathways.

In MS, one hypothesis is that insufficient peripheral IDO activity—supported by evidence of reduced IDO mRNA expression in the blood of MS patients—may fail to suppress autoreactive T cells, thereby facilitating autoimmune attacks. Conversely, within the CNS, robust IDO/KYN production during acute inflammation may contribute to ‘immune exhaustion’ over time, potentially explaining why inflammation wanes in later stages of the disease despite ongoing neurodegeneration. In addition, KP metabolites such as picolinic acid and NAD^+^ have roles in macrophage activation states and tissue repair. Depletion of NAD^+^ in active lesions (due to consumption by PARP enzymes repairing oxidative DNA damage and activation of poly(ADP-ribose) polymerase from QUIN/ROS injury) can lead to energy depletion-induced immunosuppression in situ, which may limit acute inflammation but at the expense of neuronal survival. The complex interplay between KP metabolites and immune responses is still being elucidated. However, it is clear that KP does not act in isolation, but intersects with excitotoxic, oxidative, and immune pathways to shape the course of MS.

In summary, KP activation in MS may initially represent a counter-regulatory mechanism to limit autoimmunity. However, prolonged or excessive KP flux—particularly via the KMO branch—is likely to become pathogenic as chronic accumulation of QUIN and other neurotoxins eventually outweighs any transient immunosuppressive benefits. This imbalance drives mechanisms of tissue injury—from glutamate-mediated excitotoxicity to free radical cascades—that underlie demyelination and neurodegeneration.

### 3.2. Dysregulation of the KP in MS Disease States

There is increasing evidence that the kynurenine pathway is severely dysregulated in MS and that the profile of KP metabolites differs between the relapsing-remitting and progressive phases of the disease (see Figure 2). During acute inflammatory attacks (relapses), IDO-1 is upregulated in immune cells, leading to increased conversion of TRP to KYN [49]. As a result, patients with active relapses have elevated levels of KYN and its downstream metabolites, including QUIN and 3-HK, in serum or cerebrospinal fluid compared to healthy individuals [50]. In particular, studies have reported conflicting changes in KYNA during relapse—one study found that CSF KYNA was reduced during the acute phase of relapse [51], while others observed high KYNA levels during relapse [52,53]. These discrepancies are likely to reflect differences in disease phase and time of sampling, but they highlight the dynamic nature of KP metabolite fluctuations in MS [54]. Indeed, the elevation of KYNA in some RRMS patients may represent a compensatory mechanism aimed at counteracting QUIN-induced excitotoxicity [18]. The net effect in RRMS appears to be a broad activation of the KP, producing both neurotoxic and neuroprotective outputs in response to intense inflammation [55].

As the disease enters a progressive stage (either SPMS, or PPMS), the balance of KP metabolism shifts towards a more neurotoxic profile. Prolonged inflammation appears to deplete TRP and downregulate astrocytic KYNA production, resulting in significantly lower TRP and KYNA levels in progressive MS patients. In SPMS, studies consistently find reduced KYNA in both serum and CSF, along with a relative excess of QUIN and other downstream metabolites. In PPMS (which is characterised by insidious progression from onset), most KP metabolites (KYN, 3-HK, and QUIN) are found at higher than normal levels, but KYNA levels are paradoxically low [50]. In progressive patients, a low KYNA/QUIN ratio reflects the dominance of excitotoxic and oxidative stress-promoting metabolites and correlates with more advanced disability [18]. In other words, in progressive multiple sclerosis, the protective branch of the kynurenine pathway diminishes while the potentially neurotoxic branch remains active. This results in an imbalance characterised by elevated levels of neurotoxic metabolites such as 3-HK and QUIN, and deficient production of neuroprotective KYNA. This biochemical imbalance creates a microenvironment that favours chronic neurodegeneration [56].

Therefore, the ratio between KYNA and QUIN (or KYNA and 3-HK) has become a critical indicator of the neurotoxic potential of KP at different stages of MS. These biochemical changes have important implications because they correlate closely with markers of progressive neurodegeneration. For example, CSF QUIN levels increase significantly during MS relapses and correlate directly with the extent of axonal injury. QUIN contributes to this neurodegeneration by inducing hyperphosphorylation of neurofilament proteins in neurons and astrocytes, thereby destabilising their cytoskeletal structure [9]. In support of these findings, we have previously demonstrated that CSF QUIN levels in MS patients positively correlate with neurofilament light chain (NfL) levels, a well-established biomarker of neuroaxonal damage [57]. Taken together, these observations suggest that increased KP activity, particularly through increased QUIN production, reflects and potentially drives the progression of neuronal damage in MS.

These observations suggest that acute inflammation in MS drives a surge in KP metabolism that includes transient neuroprotective responses (elevated KYNA), but with chronicity the KP tends to produce more neurotoxins that may actively contribute to remyelination failure and neurodegeneration. Monitoring KP metabolite profiles (e.g., in serum or CSF) is therefore being explored as a tool to measure disease phase and potentially predict transitions from relapsing-remitting to progressive MS. For example, a metabolomics study found that a combined panel of six markers—KYNA, QUIN, TRP, PIC, fibroblast growth factor, and TNF-α—could discriminate MS subtypes with up to ~85–91% sensitivity [58]. This highlights the biomarker potential of KP metabolites in MS.

Table 2 summarises key KP alterations reported in MS patients. Notably, despite some discrepancies among studies (owing to different sample types, disease phases, and treatments), a consistent theme is increased QUIN (and QUIN/KYNA ratio) in MS, especially during active inflammation. Such shifts in KP metabolite balance are increasingly recognised as contributors to MS pathology rather than mere bystanders.

## 4. Mitochondrial Dysfunction in MS

Mitochondrial dysfunction plays a critical role in axonal degeneration and tissue injury in multiple sclerosis [59]. Mitochondria generate ATP through oxidative phosphorylation, regulate intracellular calcium, and mediate apoptosis. In MS, chronic demyelination significantly increases the energy demand of axons due to reduced conduction efficiency and simultaneously impairs mitochondrial energy production under inflammatory conditions, leading to axonal energy failure.

Post-mortem analysis of MS tissue reveals numerous mitochondrial abnormalities, including increased mitochondrial DNA (mtDNA) mutations, respiratory chain enzyme deficiencies, altered mitochondrial gene expression, and structural damage. In particular, chronic MS lesions often show mtDNA deletions and reduced expression of the respiratory complex components COX1 and ND1. Histochemically, demyelinated axons often show focal cytochrome c oxidase (complex IV) deficiency and compensatory enzyme upregulation (e.g., increased succinate dehydrogenase activity) to maintain ATP synthesis [29]. In addition, a mosaic pattern of COX-deficient and normal axons suggests clonal expansion of mtDNA mutations, similar to mitochondrial diseases [60]. Electron microscopy also reveals swollen, vacuolated mitochondria with disrupted cristae and activation of the mitochondrial permeability transition pore (PTP), indicating severe mitochondrial damage [61].

### Role of KP Metabolites in Mitochondrial Dysfunction in MS

The neurotoxic KP metabolites not only cause immediate excitotoxic and oxidative damage, but also directly impair mitochondrial function—and thus cellular energy metabolism—through mechanisms such as calcium overload. As described above, QUIN-driven NMDA receptor overactivation leads to excessive Ca^2+^ influx into neurons and oligodendrocytes. Mitochondria take up calcium as a buffering response, but massive Ca^2+^ overload can trigger mitochondrial permeability transition, swelling, and release of pro-apoptotic factors. Elevated intra-mitochondrial Ca^2+^ also disrupts the electron transport chain, increasing mitochondrial ROS production and further reducing ATP output [9]. This cascade links excitotoxicity to mitochondrial damage: in MS lesions, areas of high QUIN and glutamate exposure are likely to show focal mitochondrial depolarisation and energy failure.

In addition to its excitotoxic effects, QUIN exerts direct toxic effects on mitochondrial function. Experimental evidence shows that QUIN can inhibit key mitochondrial enzymes, including monoamine oxidase-B (MAO-B), and disrupt metabolic pathways such as the Krebs cycle and gluconeogenesis. These effects occur independently of NMDA receptor activation, suggesting that QUIN has intrinsic mitochondrial targets. Indeed, QUIN exposure leads to progressive mitochondrial dysfunction even in the absence of excitotoxic signalling. In neuronal models, QUIN accumulation leads to reduced activity of respiratory chain complexes, loss of mitochondrial membrane potential, and cytochrome c release—hallmarks of mitochondria-mediated apoptosis. This cascade subsequently activates caspases and contributes to cytoskeletal degradation, ultimately forcing both neurons and glial cells into an energy-depleted, apoptotic state [33].

Similarly, 3-hydroxykynurenine (3-HK), another neurotoxic metabolite of the kynurenine pathway, has been shown to impair mitochondrial function. At elevated concentrations, 3-HK decreases intracellular NAD^+^ levels and mitochondrial membrane potential in cultured glial cells [62]. The decrease in NAD^+^ reflects its consumption during 3-HK-induced oxidative reactions or during PARP-mediated DNA repair responses to oxidative stress. As NAD^+^ is essential for mitochondrial ATP production, its depletion triggers a metabolic crisis culminating in cellular energy failure. This is evidenced by increased release of lactate dehydrogenase (LDH), a marker of membrane breakdown and cell death under conditions of mitochondrial dysfunction and energy depletion [62,63].

Taken together, these findings underscore that kynurenine pathway metabolites such as QUIN and 3-HK contribute to MS pathology not only through excitotoxicity and oxidative stress, but also by directly disrupting mitochondrial integrity and cellular energy metabolism.

Another important consideration is that although the KP ultimately culminates in NAD^+^ synthesis, its overactivation can paradoxically lead to local NAD^+^ depletion. In the context of MS, if QUIN accumulates faster than it can be converted to NAD^+^ by QPRT—as may occur in microglia, which actively produce QUIN but express limited QPRT—then excess QUIN will persist and exert ongoing neurotoxic effects, rather than being neutralised by NAD^+^ synthesis. In addition, QUIN-induced activation of poly(ADP-ribose) polymerase (PARP) as part of the DNA damage response depletes intracellular NAD^+^ pools [64]. This creates a vicious cycle in which KP activity theoretically generates NAD^+^, but downstream QUIN-driven processes deplete it. The net result, particularly in chronic MS lesions, may be a functional NAD^+^ deficit within neurons and glia, exacerbating cellular energy failure. As NAD^+^ is essential for cellular survival under oxidative stress, its depletion contributes significantly to neurodegeneration.

The downstream consequences of KP-mediated mitochondrial dysfunction are far-reaching. Neurons deprived of sufficient energy conduct impulses inefficiently and are more prone to degeneration. Axons with damaged mitochondria undergo Wallerian degeneration, while the neuronal soma may atrophy or undergo apoptosis [4]. Oligodendrocyte precursor cells (OPCs), which require substantial energy for differentiation and remyelination, may fail to mature in a low-ATP environment, impairing endogenous repair mechanisms [65]. Over time, this persistent energy crisis at the cellular level manifests clinically as progressive neurological disability: as more axons and neurons lose function or die, irreversible deficits accumulate. This concept is consistent with the notion that progressive MS is characterised by a “slow burn” of neurodegeneration driven by metabolic injury rather than active inflammation. Evidence from neuropathological studies supports this metabolic-inflammatory interplay. Chronically active MS lesions often show a rim of activated microglia—major sources of QUIN and ROS—surrounding a demyelinated core devoid of viable axons [3]. Within surviving axons, mitochondrial density is often increased as a compensatory response; however, many of these mitochondria appear abnormally swollen or depolarised [66]. Elevated levels of oxidative damage markers [3] and downregulated expression of mitochondrial respiratory chain subunits [67] further highlight the mitochondrial dysfunction present in these lesions. KP likely contributes to these pathologies by providing a molecular link between chronic inflammation and cellular metabolic failure.

Interrupting this pathological cycle is therefore an important therapeutic goal. Reducing KP-mediated mitochondrial damage could preserve neuronal and glial viability in the face of chronic inflammation, potentially delaying or preventing the transition to progressive MS. Preclinical studies have shown that inhibition of KMO—a key enzyme that drives the production of 3-HK and QUIN—has neuroprotective effects, in part by preserving mitochondrial function under inflammatory conditions [37]. Similarly, therapeutic strategies aimed at supporting mitochondrial health—such as antioxidant treatments or metabolic enhancers—may mitigate the deleterious effects of toxic KP metabolites.

The bidirectional relationship between KP dysregulation and mitochondrial dysfunction suggests that interventions targeting one may benefit the other: protecting mitochondria may reduce the effects of neurotoxic kynurenines, while modulating KP may help maintain mitochondrial integrity. Notably, some KP-targeted therapies, such as laquinimod (an oral MS drug candidate), have demonstrated mitochondrial-protective effects in experimental models—likely through modulation of microglial activation and suppression of QUIN production.

## 5. Pharmacological Modulation of the KP in MS

### 5.1. Introduction to Pharmacological Modulation

Given the increasing evidence that dysregulated KP metabolism and mitochondrial dysfunction are central drivers of neurodegeneration in multiple sclerosis, there is growing interest in therapeutic strategies that directly target these mechanisms. To date, most approved MS therapies are immunomodulatory or immunosuppressive agents designed to limit peripheral immune activity. While these disease-modifying therapies (DMTs) are effective in reducing relapse frequency in RRMS, they have limited efficacy in halting the slow, progressive neurodegeneration that characterises SPMS and PPMS. This therapeutic gap highlights the urgent need for neuroprotective strategies that act within the CNS to preserve neuronal and glial integrity.

One promising avenue is the development of drugs that either modulate KP or enhance mitochondrial resilience. Such interventions aim to neutralise neurotoxic kynurenine metabolites (e.g., QUIN and 3-HK), increase NAD^+^ availability, mitigate oxidative stress, and support mitochondrial function—all of which may slow or prevent neurodegeneration. In particular, drug repositioning—the repurposing of existing compounds with established safety profiles for new indications—has emerged as an effective strategy for accelerating therapeutic development. This approach is particularly attractive for progressive MS, where treatment options remain limited and time to clinic is critical.

A key concept in therapeutic development for MS is that combining anti-inflammatory activity with direct neuroprotection could synergistically slow the progression of disability. Targeting the KP is attractive because it could theoretically do both: inhibiting pro-inflammatory enzymes such as IDO or KMO could not only reduce the production of neurotoxic metabolites but also dampen immune activation (since IDO activity influences T-cell responses). Conversely, increasing levels of a neuroprotective metabolite such as KYNA could protect neurons and oligodendrocytes from excitotoxic and oxidative injury, even in the presence of ongoing inflammation. In addition, approaches that support mitochondrial function (e.g., by increasing NAD^+^ availability or reducing oxidative stress) may help to break the cycle of energy failure in MS. Ultimately, the integration of KP- and mitochondria-targeted therapies with existing immunomodulatory treatments may offer a more comprehensive approach to MS treatment—one that addresses both the inflammatory and degenerative components of the disease.

The following sections discuss specific pharmacological strategies and agents—some investigational, some already available—that exemplify these principles. We first examine laquinimod, a drug developed specifically for MS that has KP-modulating properties, and then review other repurposed drugs (such as alpha-lipoic acid and metformin) that are being investigated for their ability to influence KP and mitochondrial dysfunction in MS.

### 5.2. Laquinimod as a KP Modulator

Laquinimod is an oral immunomodulatory compound that has been investigated as a disease-modifying therapy in MS. Chemically, laquinimod (N-ethyl-N-phenyl-5-chloro-1,2-dihydro-4-hydroxy-1-methyl-2-oxo-3-quinolinecarboxamide) is a derivative of roquinimex and bears a structural similarity to kynurenine metabolites (indeed, its quinoline-carboxamide core is similar to that of KYNA, see Figure 3) [68,69]. This structural insight initially suggested that laquinimod might interact with KP or its receptors [70]. Although the exact mechanism of action of laquinimod is not fully understood, it has demonstrated a dual profile of immunomodulation and neuroprotection. Unlike classical immunosuppressants, laquinimod does not broadly suppress lymphocyte proliferation, but rather shifts the immune response towards an anti-inflammatory state, while also acting within the CNS to promote neuroprotective pathways [71].

In vitro and animal studies have shown that laquinimod reduces antigen presentation and inflammatory cell migration. It downregulates the expression of MHC class II and co-stimulatory molecules on antigen-presenting cells and increases regulatory populations of IL-10 secreting B cells [72]. Laquinimod-treated immune cells produce less pro-inflammatory cytokines such as IFN-γ, IL-17, GM-CSF, and TNF-α, and more anti-inflammatory IL-4, indicating a shift from a Th1/Th17-dominated response to a Th2/regulatory profile. In EAE, laquinimod reduced CNS inflammation: it reduced monocyte and T-cell infiltration by decreasing adhesion molecule expression (e.g., VLA-4) and matrix metalloproteinase-9 activity [8]. Notably, laquinimod also limited microglial activation in these models, resulting in less axonal damage [70]. Perhaps most interestingly, laquinimod promoted remyelination in animal studies appeared to protect oligodendrocyte progenitor cells and support their maturation during the repair phase of lesions [73].

In addition to these immune effects, laquinimod has been shown to increase levels of brain-derived neurotrophic factor (BDNF) in the CNS, a key neurotrophin for the survival of neurons and oligodendrocytes [74]. In an EAE study, laquinimod treatment led to increased BDNF expression in several brain regions and was associated with reduced axonal loss and astrogliosis [75]. Similarly, in a clinical trial of MS patients, those treated with laquinimod had significantly higher serum BDNF levels than those on placebo [76]. This suggests that laquinimod may exert neuroprotective effects independent of its immune modulation—possibly through direct interaction with CNS cells or pathways (whether via KP modulation or other mechanisms such as aryl hydrocarbon receptor activation remains to be investigated [77]).

Laquinimod has undergone phase III trials in RRMS (the ALLEGRO and BRAVO trials), which have provided some insight into its efficacy. In ALLEGRO, a 2-year placebo-controlled study, laquinimod (0.6 mg daily) significantly but modestly reduced the annualised relapse rate compared to placebo. It also significantly reduced the risk of confirmed disability progression by approximately 40% compared to placebo. MRI results showed that laquinimod reduced the number of new or enlarged T2 lesions and gadolinium-enhancing lesions and, importantly, slowed brain atrophy: the rate of brain volume loss was lower in the laquinimod group than in the placebo group [78]. The parallel BRAVO study, which included an active comparator (interferon-β) plus placebo, showed a more modest effect on relapse rate and did not reach significance for some MRI endpoints, although brain atrophy was again significantly reduced with laquinimod [79]. The overall safety profile in these trials was favourable. However, in preclinical animal studies, chronic laquinimod exposure was associated with an increased incidence of malignancies (mainly rodent cancers). Although there was no cancer signal in human studies, regulators were concerned about this potential risk [71]. This, combined with its relatively modest efficacy (particularly in relapsing forms, where it was less effective than first-line injectables), led to laquinimod not being approved for MS in Europe or the US.

Despite this setback, laquinimod remains a proof-of-concept that targeting KP and related pathways can provide both anti-inflammatory and neuroprotective benefits in MS. Its ability to increase BDNF and potentially promote remyelination is particularly noteworthy and has stimulated interest in developing next-generation compounds that retain these neuroprotective properties with greater efficacy or safety. Laquinimod’s structural relationship with kynurenines also raises interesting questions about whether it can bind to receptors such as AHR or inhibit enzymes such as KMO—areas of ongoing research [80].

In summary, laquinimod has demonstrated that a small molecule can modulate the immune response in MS without broad immunosuppression and, perhaps more importantly, promote a more repair-friendly environment in the CNS. These lessons are being applied as researchers explore other pharmacological modulators of the kynurenine pathway and mitochondrial function in MS. Notably, in January 2025, the United States Patent and Trademark Office granted Active Biotech a patent (US 12,208,091) for the use of laquinimod in the treatment of eye diseases associated with excessive vascularisation, such as wet age-related macular degeneration, proliferative diabetic retinopathy, and ischaemic retinopathy, expanding its potential therapeutic applications beyond neuroinflammatory disorders.

### 5.3. Drug Repurposing for MS Treatment: Targeting KP and Mitochondria

The strategy of drug repurposing has brought several existing drugs into the spotlight as potential MS therapies, addressing both metabolic and neurodegenerative aspects of the disease. Two particularly promising candidates identified through systematic analysis are R-α-lipoic acid (ALA) and metformin, both of which have shown the ability to attenuate mechanisms associated with KP and mitochondrial dysfunction. In addition, a number of other drugs—some already approved for other indications—have mechanistic rationales or preliminary evidence suggesting that they may modulate KP or protect against its downstream damage in MS. These candidates and the rationale for their use in MS are discussed below.

ALA is an endogenous antioxidant and cofactor for mitochondrial enzymes (particularly pyruvate dehydrogenase) that has shown beneficial effects in MS models. Lipoic acid can penetrate the CNS and has dual effects as an immunomodulator and antioxidant. In EAE, ALA supplementation consistently reduced disease severity, which was associated with fewer inflammatory cells infiltrating the CNS and less oxidative damage. ALA appears to strengthen the blood–brain barrier, reduce immune cell migration by downregulating adhesion molecules, and protect the integrity of endothelial tight junctions. Within the CNS, ALA inhibits the activity of T cells and microglia, leading to reduced production of TNF-α and IFN-γ, and neutralises ROS and reactive nitrogen species such as NO. These effects counteract the pro-oxidant and pro-inflammatory environment created by KP dysregulation in MS. In addition, ALA’s antioxidant activity may reduce the oxidative stress that fuels mitochondrial dysfunction [5].

Clinically, ALA has shown promise in a phase II trial in SPMS. In a two-year randomised controlled trial in patients with secondary progressive MS, high-dose oral lipoic acid (1200 mg daily) significantly slowed whole-brain atrophy compared with placebo. The rate of brain volume loss in the ALA group was only about one-third that of the placebo group (annualised percentage change in brain volume of −0.21% vs. −0.65%, *p* = 0.002), representing a 68% reduction in the rate of brain atrophy. This is a remarkable effect size on a key measure of neurodegeneration. There were also trends towards clinical benefit in the ALA-treated group—for example, a slower timed 25-foot walk—although the study was not powered for disability outcomes. ALA was generally well tolerated, with a favourable safety profile apart from some gastrointestinal adverse events (and a few isolated reversible renal events in the treatment group) [81]. These results provide Class I evidence that ALA may have neuroprotective effects in MS. The exact mechanisms in humans are still being elucidated, but given ALA’s broad antioxidant capacity and its effect on immune trafficking, it is likely to reduce the burden of CNS-mediated damage. Indeed, lipoic acid’s role as a mitochondrial cofactor may increase pyruvate dehydrogenase activity and improve aerobic metabolism in cells under inflammatory stress. By supporting mitochondrial energy production and directly scavenging free radicals, ALA addresses two important downstream consequences of KP activation. Larger trials of ALA are underway (including one arm of the innovative multi-arm OCTOPUS study in progressive MS in the UK) to confirm these findings and potentially bring ALA into routine use as an adjunctive neuroprotective therapy.

Metformin: Metformin is a widely used oral drug for type 2 diabetes that activates AMPK (AMP-activated protein kinase) and modulates cellular metabolism. Interest in metformin for MS arose from observations that it can reprogramme metabolism in a way that promotes tissue repair and reduces inflammation. Metformin has been shown to reverse age-related regenerative failure in the CNS: in aged animals, metformin ‘rejuvenated’ OPCs and restored their ability to remyelinate lesions. In addition, metformin exerts anti-inflammatory effects by inhibiting NF-κB and IL-17 signalling and reducing microglial activation. In the EAE model, metformin treatment resulted in milder disease—mice treated early with metformin had improved motor outcomes, less spinal cord demyelination, and fewer activated microglia than untreated EAE controls [82]. These benefits were seen when metformin was started at disease induction (prophylactically), highlighting the drug’s ability to limit initial immune-mediated damage. Metformin’s mechanisms in EAE included a reduction in pro-inflammatory cytokines and an increase in oligodendroglial lineage cells, suggesting both immunoregulatory and pro-myelinating effects.

An intriguing link between metformin and KP is that metformin may down-regulate KP under certain conditions. A clinical trial in individuals with insulin resistance found that successful metformin treatment was associated with down-regulation of kynurenine pathway activity (as reflected by reduced KYN/TRP ratios and metabolite levels) [83]. This suggests that metformin may indirectly inhibit IDO or related enzymes, perhaps through its effects on systemic inflammation and insulin signalling. In the CNS, metformin’s activation of AMPK promotes cellular resilience to stress—for example, it may induce autophagy and mitochondrial biogenesis, potentially offsetting mitochondrial dysfunction in MS lesions. There is also evidence that metformin can increase NAD^+^ levels by altering the NAD^+^/NADH ratio and upregulating NAD^+^ synthesis pathways [84], which could counteract the NAD^+^ depletion caused by chronic KP activation.

These multiple effects make metformin a strong candidate in progressive MS, where conventional anti-inflammatory drugs alone have minimal impact on disability progression. A multi-centre phase II study (MACSiMiSE) is currently underway to test metformin in non-active progressive MS. Researchers are investigating whether two years of metformin (up to 2550 mg/day) can delay the progression of walking disability and brain atrophy compared with placebo. The trial will also include advanced MRI scans to assess structural preservation and potential remyelination [85]. If successful, metformin—an inexpensive and well-known drug—could be repurposed to fill the treatment gap in progressive MS. Its safety profile is well characterised; apart from gastrointestinal side effects and very rare lactic acidosis in predisposed patients, metformin is generally well tolerated. Notably, metformin may be synergistic with remyelinating therapies: a separate trial is investigating metformin in combination with the antihistamine clemastine (which promotes OPC differentiation) to see if the combination improves myelin repair in MS [85]. In summary, metformin is a promising agent that attacks MS pathology on multiple fronts—calming inflammatory KP activation, supporting mitochondrial health, and rejuvenating endogenous repair mechanisms in the CNS.

In addition to ALA and metformin, several other drugs have mechanisms that overlap with KP or its downstream effects. Table 3 summarises six drugs that have been identified as promising candidates for further investigation in MS: ibudilast, riluzole, amiloride, pirfenidone, fluoxetine, and oxcarbazepine. These drugs are already approved for other indications, but have shown neuroprotective or anti-inflammatory properties that may be of benefit to people with MS.

## 6. Conclusions

Nevertheless, a critical limitation inherent in targeting the KP is the “Janus-faced” nature of its metabolites. This dualistic property implies that therapeutic modulation requires precise calibration to avoid disrupting essential physiological functions such as immune tolerance, energy metabolism, and cellular homeostasis. Moreover, MS pathogenesis involves additional molecular players beyond KP metabolites, including various cytokines, chemokines, and other inflammatory mediators, as well as genetic and environmental factors that also significantly influence disease trajectory. Consequently, focusing solely on KP metabolites might oversimplify the intricate pathogenic network of MS.

Another major limitation of current knowledge is the incomplete understanding of how KP metabolism dynamically interacts with other metabolic and signalling pathways in the CNS microenvironment. The temporal and spatial complexity of KP metabolite production within different cell populations in the CNS—such as astrocytes, microglia, neurons, and infiltrating immune cells—adds another layer of complexity. Thus, therapeutic interventions will require precision not only in metabolite selection, but also in timing and localisation.

Mitochondrial dysfunction complicates therapeutic strategies because it is both a downstream consequence and a central driver of MS pathophysiology. Chronic oxidative stress and excitotoxic calcium overload prevent axons from meeting their metabolic demands, resulting in cumulative damage over time. Any strategy aimed at preserving mitochondria must strike a delicate balance between promoting cell survival and avoiding exacerbation of chronic inflammation or other metabolic disorders.

Future therapeutic interventions will need to adopt nuanced, multifaceted approaches that selectively inhibit neurotoxic pathways (e.g., KMO) while preserving or enhancing protective metabolites such as KYNA. Furthermore, therapeutic modulation of the kynurenine pathway must take into account individual patient variability due to genetic and environmental heterogeneity, which poses an additional challenge to the development of generalisable strategies. Combining these strategies with mitochondrial support and broader immunomodulatory therapies may maximise neuroprotection in MS. Recognising the complexity and limitations of targeting KP, together with a thorough exploration of other contributing molecules and pathways, will be critical to translating theoretical advances into meaningful clinical outcomes.

Future studies and clinical trials should aim to address these limitations by adopting a comprehensive biomarker-guided approach, integrating multiple molecular targets, and optimising individualised treatment protocols. Multifaceted therapies combining selective modulation of neurotoxic and neuroprotective KP metabolites with mitochondrial support and broader immunomodulatory strategies are likely to be essential to achieve significant and durable clinical benefits in MS.

Ultimately, targeting the kynurenine pathway and mitochondrial dysfunction offers a promising two-pronged approach to suppress immune-mediated inflammation and protect the nervous system from damage. Ongoing and future clinical trials will determine the extent to which these theoretical benefits can be realised in practice. There is cautious optimism that integrating metabolic and neuroprotective therapies with immunotherapies could significantly alter the course of MS, potentially slowing, halting or even partially reversing disability. Interdisciplinary research, as exemplified by studies of the kynurenine pathway, is providing clinicians with new tools to monitor and treat MS, offering hope for significantly improved patient outcomes.

## Figures and Tables

**Figure 1 ijms-26-05098-f001:**
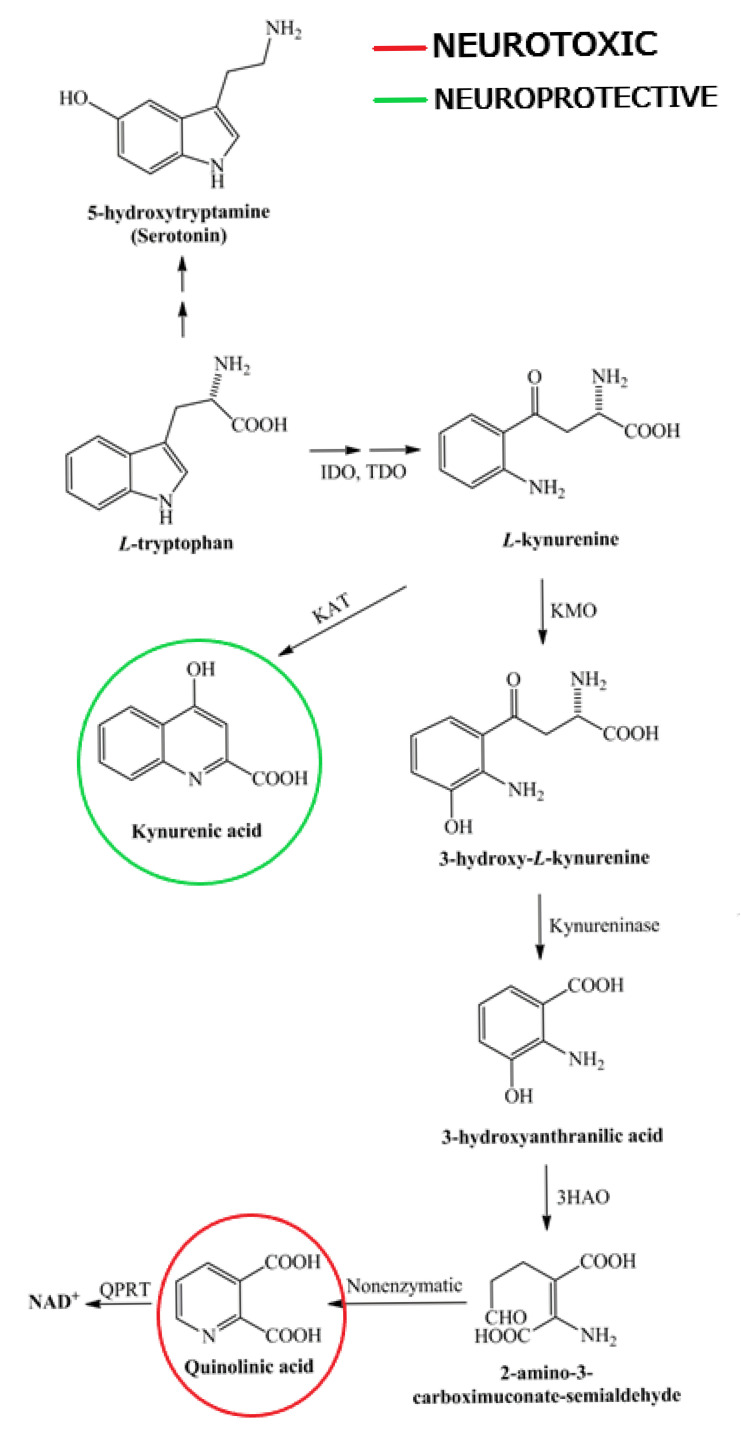
Scheme of the kynurenine pathway. Tryptophan, an essential amino acid, is primarily broken down through the kynurenine pathway, resulting in the formation of both neurotoxic and neuroprotective metabolites. 3-HAO—3-hydroxyanthranilate oxidase; IDO—indoleamine 2,3-dioxygenase; KAT—kynurenine aminotransferase; KMO—kynurenine 3-monooxygenase; NAD^+^—nicotinamide adenine dinucleotide; QPRT—quinolinate phosphoribosyltransferase; TDO—tryptophan 2,3-dioxygenase.

**Figure 2 ijms-26-05098-f002:**
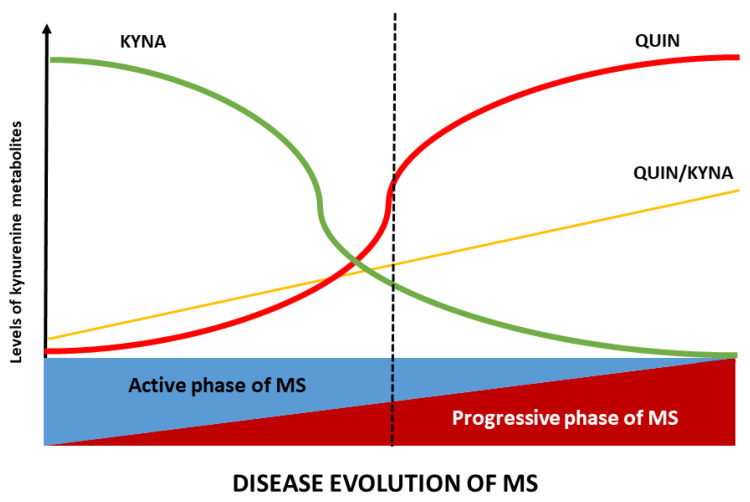
Alterations of the KP profile during the course of MS: the metabolic profile of KP varies according to the stage of MS. In the active phase, kynurenic acid (KYNA) levels are elevated, whereas in the progressive phase, quinolinic acid (QUIN) levels and the QUIN/KYNA ratio increase. These metabolic changes in MS are closely related to disease activity. Note: the QUIN/KYNA ratio exhibits a notable similarity to PIRA, suggesting its potential as a biomarker for neurodegeneration in MS. Abbreviations: KYNA, kynurenic acid; QUIN, quinolinic acid; MS, multiple sclerosis.

**Figure 3 ijms-26-05098-f003:**
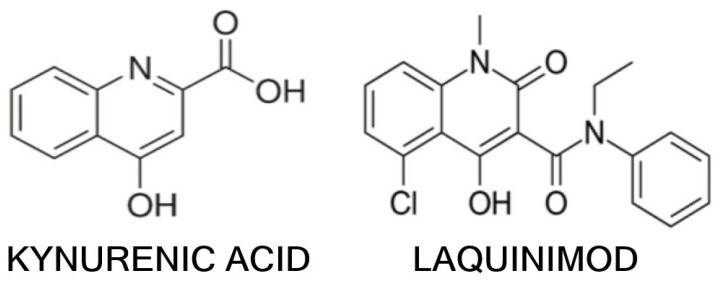
The chemical structure of kynurenic acid and laquinimod.

**Table 1 ijms-26-05098-t001:** The metabolites of the kynurenine pathway significantly influence the progression of MS. Some metabolites (such as QUIN and 3-HK) exacerbate the disease by directly damaging brain cells, whereas KYNA protects cells by limiting damage. KYN itself plays a dual role depending on the context, initially suppressing harmful immune activity but later indirectly contributing to neurotoxicity. PIC adds a beneficial dimension to KP by counteracting neurotoxic effects through metal chelation and anti-excitotoxic activity.

Metabolite	Primary Source	Main Actions in MS	Effect
**Quinolinic acid (QUIN)**	Activated microglia and macrophages	- **Excitotoxicity:** Direct NMDA receptor activation, neuronal Ca^2+^ overload → neuronal death. - **Glutamate dysregulation:** Increases glutamate release, reduces uptake, and damages oligodendrocytes. - **Oxidative stress:** Enhances ROS production, damages myelin, and DNA. - **Energy failure:** Inhibits mitochondrial function, depletes NAD^+^. - **Cytoskeletal disruption:** Tau hyperphosphorylation, impairs neuronal integrity.	🚨 **Neurotoxic**
**Kynurenic acid (KYNA)**	Astrocytes	- **Neuroprotection:** NMDA receptor antagonist, reduces excitotoxicity. - **Antioxidant:** Neutralises reactive species, protects against oxidative damage. - **Neuromodulation:** Suppresses excessive glutamate release. - Generally insufficient in MS lesions, potential therapeutic target.	✅ **Neuroprotective**
**3-Hydroxykynurenine (3-HK)**	Microglial cells	- **Oxidative damage:** Generates ROS (hydrogen peroxide, hydroxyl radicals). - **Synergistic neurotoxicity with QUIN:** Amplifies neuronal and oligodendrocyte damage when combined with QUIN.	🚨 **Neurotoxic**
**Kynurenine (KYN)**	IDO-expressing immune cells	- **Immune regulation:** Modulates T-cell responses, can induce T-cell apoptosis, and promote immune tolerance via AHR activation. - **Dual role:** Protective initially (immune suppression), harmful long-term due to metabolite conversion to QUIN and 3-HK.	⚠ **Context-dependent (Immune modulation, potentially harmful)**
**Picolinic acid (PIC)**	Derived from ACMSD-expressing cells (neurons, astrocytes, and macrophages)	- **Metal chelation:** Reduces oxidative damage by binding neurotoxic metals (Fe^2+^, Zn^2+^). - **Anti-excitotoxic:** Inhibits QUIN-induced neuronal damage.	✅ **Neuroprotective**

Legend: 🚨 = Harmful metabolite promoting MS disease progression. ✅ = Beneficial, protective metabolite that counters harmful effects. ⚠ = Dual role; effects depend on concentration and context.

**Table 2 ijms-26-05098-t002:** Summary of KP metabolite patterns in different MS phenotypes.

MS Phenotype	Key KP Metabolic Features	Clinical/Pathological Correlation	References
RRMS	Surges in KYN and QUIN during relapses; some compensatory increase in KYNA during recovery	Acute inflammatory relapses with partial remission	See review Fathi et al. [43]
SPMS	Chronic elevation of QUIN, reduced KYNA, depleted TRP	Persistent neurodegeneration, fewer apparent relapses	[50] and see review Fathi et al. [43]
PPMS	High QUIN and 3-HK from early stages, low KYNA	Continuous progression with minimal inflammatory bursts	[18]

**Table 3 ijms-26-05098-t003:** Repurposed drugs for MS targeting neuroprotection.

Drug	Original Use	Mechanism in MS Context	Clinical Evidence in MS
Ibudilast	Asthma (Japan)	PDE inhibition, ↓ microglial activation	Slow down the rate of brain atrophy [86]. Reduces slowly enlarging lesions in progressive MS [87]
Riluzole	Amyotrophic lateral sclerosis (ALS)	↓ Glutamate release, counters excitotoxicity	A pilot study of riluzole in progressive MS showed reduced cervical cord atrophy and fewer new brain T1 hypointense lesions [88], but another study in early RRMS or CIS found no reduction in atrophy rates [89].
Amiloride	Potassium-sparing diuretic	Blocks ASIC1, prevents Ca^2+^ overload in axons	A pilot study in individuals with progressive multiple sclerosis demonstrated a significant reduction in whole-brain atrophy [90].
Pirfenidone	Pulmonary fibrosis	Minimise demyelination by inhibiting the production and/or action of TNF-alpha	Pirfenidone may have a significant impact on clinical disability and bladder function in patients with SPMS [91].
Fluoxetine	Antidepressant (SSRI)	Anti-inflammatory in microglia, possible IDO suppression	In a small, inconclusive trial, fluoxetine was associated with modest but statistically non-significant improvements in some clinical progression markers [92].
N-acetyl cysteine	Acetaminophen-induced hepatotoxicity	Glutathione precursor with antioxidant properties	No large MS study; Decreased lipid peroxidation and improved anxiety symptoms in MS patients [93].

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
