# Peer review of "Kynurenines and Mitochondrial Disturbances in Multiple Sclerosis"

_ijms, 2025, doi:10.3390/ijms26115098_

Round 1

Reviewer 1 Report

Comments and Suggestions for Authors

In this review, the authors provide a comprehensive overview of the kynurenine pathway and its metabolites and how their disruption, together with mitochondrial dysfunction, contributes to the mechanisms of multiple sclerosis pathology. It also discusses novel therapeutic approaches that modulate the kynurenine pathway or ameliorate mitochondrial dysfunction in MS.

It is a well-presented review that covers a very interesting topic. The review seems solid and comprehensive and leads to some consistent conclusions. However, some aspects need clarification and explanation or corrections.

Comments

1- The abbreviation QUIN is not explained as quinolinic acid in the manuscript, but only in the asbtract and in Figure 2, but not in the text where it is used.

2- Line 169. QUIN could be the subtitle of 3.1.1, with the full name as: Quinolinic acid (QUIN), as, KYNA as 3.1.2 be: kynurenic acid (KYNA), 3.1.3 be 3-HK as: 3-hydroxykynurenine (3-HK), and 3.1.4 be picolinic acid (PIC).

3- In 3.1, there seems to be a lack of information about picolinic acid in multiple sclerosis (only line 300 mentions it), but there is some work linking this compound to MS. Insert a short paragraph as for the other metabolites (rewrite lines 294-303).

4- Line 405. is this title a subtitle? It is 4.1?

5- A figure summarizing section 3.1 and showing a summarized list of the effects of the metabolites would be interesting for general readers. It shows the four metabolites and outlines their effects in relation to MS. It also indicates which of them are harmful and which are neuroprotective.

6- It would be desirable to include references in Table 1. (Shortening the width of columns 2 and 3).

7- It would be important to point out the limitations of the study in the discussion.

Author Response

Dear Reviewer 1,

Thank you for your valuable time and effort to improve the content of our manuscript entitled „Kynurenines and mitochondrial disturbances in Multiple Sclerosis”. We have carefully revised the manuscript according to the recommendations, and below we provide our point-by-point answers to you. We have made the changes directly to the manuscript, as well as highlighted the changes below.

Please find our answers below:

1: We would like to clarify that the full name quinolinic acid (QUIN) was spelled out at first mention in the main text on page 3, line 119. Following standard scientific writing conventions, we subsequently used the abbreviation QUIN throughout the manuscript for readability and consistency. Nevertheless, we will be happy to rephrase or reintroduce the full name at a later point in the text if the reviewer or editor deems it necessary for clarity. Changes are highlighted in yellow.

2: In accordance with the recommendation, we have revised the subtitles of subsections 3.1.1–3.1.4 to include both the full names and abbreviations of the respective kynurenine pathway metabolites. Changes are highlighted in yellow. The new headings now read:

  • 3.1.1 Quinolinic acid (QUIN)

  • 3.1.2 Kynurenic acid (KYNA)

  • 3.1.3 3-Hydroxykynurenine (3-HK)

  • 3.1.4 Picolinic acid (PIC)

3: In response, we have expanded the discussion of picolinic acid and created a dedicated subsection titled 3.1.4 Picolinic acid (PIC). In this section, we summarize current evidence on PIC’s neuroprotective roles in multiple sclerosis, including its metal-chelating properties, ability to counteract quinolinic acid-induced excitotoxicity, and influence on macrophage phenotype. 

4: This subtitle has been corrected.

5: In response, we have created Table 1, which outlines the four key kynurenine pathway metabolites discussed (QUIN, KYNA, 3-HK, and PIC), along with their known effects in multiple sclerosis. The table highlights whether each metabolite is primarily neurotoxic or neuroprotective and briefly summarizes their main mechanisms of action.

6: We have revised the table accordingly and renamed it Table 2. Column widths have been adjusted to improve readability, and appropriate literature references have been added. We agree that these modifications enhance the scientific rigor and clarity of the table.

7: In response, we have completely rewritten the Conclusion section to incorporate a more balanced and comprehensive summary of the study's findings. The revised version now explicitly addresses the limitations.

Grammar issues and abbrevations were corrected.

We thank again for the valuable and constructive comments of the Reviewers which have highly improved the quality of the manuscript. We hope that the manuscript will be accepted for publication.

Kind regards,

The Authors

Reviewer 2 Report

Comments and Suggestions for Authors These are my answers:   • What is the main question addressed by the research? - The kynurenine pathway (KP) and mitochondrial dysfunction are central to the pathogenesis of MS, linking immune-mediated attack with CNS tissue degeneration.
  • Targeting these pathways offers a two-pronged approach to MS treatment: quelling the 721 immune-driven fire and fortifying the nervous system against damage.
KP activation, driven by inflammatory cytokines, leads to the production of both 16 neuroprotective (e.g., kynurenic acid, KYNA) and neurotoxic (e.g., quinolinic acid, QUIN) 17 metabolites. Imbalance between these metabolites, particularly increased QUIN production, exacerbates glutamate excitotoxicity, oxidative stress, and mitochondrial dysfunction, contributing to neuronal and oligodendrocyte damage.  Mitochondrial dysfunction plays a critical role in the pathophysiology of MS, exacerbating neurodegeneration through impaired energy metabolism and 21 oxidative stress.  This review integrates current understanding of KP dysregulation in multiple sclerosis across disease stages. • Do you consider the topic original or relevant to the field? Yes. Does it address a specific gap in the field? This review integrates current understanding of KP dysregulation in multiple sclerosis across disease stages. • What does it add to the subject area compared with other published material? An updated review about Kynurenines and mitochondrial disturbances in Multiple Sclerosis
• What specific improvements should the authors consider regarding the methodology? Nothing. It is perfect. 
• Are the conclusions consistent with the evidence and arguments presented and do they address the main question posed? Yes. An updated review about Kynurenines and mitochondrial disturbances in Multiple Sclerosis
• Are the references appropriate? Yes
• Any additional comments on the tables and figures. No

Congratulations

Author Response

Dear Reviewer 2,

Thank you for your valuable time and effort to improve the content of our manuscript entitled „Kynurenines and mitochondrial disturbances in Multiple Sclerosis”.

We thank again for the valuable and constructive comments of the Reviewers which have highly improved the quality of the manuscript. We hope that the manuscript will be accepted for publication.

Kind regards,

The Authors

Round 2

Reviewer 1 Report

Comments and Suggestions for Authors

The authors have addressed all of the questions and comments, consequently the manuscript has been improved in precision.